# Genetic and Genome-Wide Association Analysis of Yearling Weight Gain in Israel Holstein Dairy Calves

**DOI:** 10.3390/genes12050708

**Published:** 2021-05-10

**Authors:** Moran Gershoni, Joel Ira Weller, Ephraim Ezra

**Affiliations:** 1Institute of Animal Sciences, ARO, The Volcani Center, Bet Dagan 50250, Israel; gmoran@volcani.agri.gov.il; 2Israel Cattle Breeders Association, Caesaria Industrial Park, Caesaria 38900, Israel; ephraim@icba.co.il

**Keywords:** dairy cattle, growth rate, animal model, genetic analysis, genomic analysis

## Abstract

Yearling weight gain in male and female Israeli Holstein calves, defined as 365 × ((weight − 35)/age at weight) + 35, was analyzed from 814,729 records on 368,255 animals from 740 herds recorded between 1994 and 2021. The variance components were calculated based on valid records from 2008 through 2017 for each sex separately and both sexes jointly by a single-trait individual animal model analysis, which accounted for repeat records on animals. The analysis model also included the square root, linear, and quadratic effects of age at weight. Heritability and repeatability were 0.35 and 0.71 in the analysis of both sexes and similar in the single sex analyses. The regression of yearling weight gain on birth date in the complete data set was −0.96 kg/year. The complete data set was also analyzed by the same model as the variance component analysis, including both sexes and accounting for differing variance components for each sex. The genetic trend for yearling weight gain, including both sexes, was 1.02 kg/year. Genetic evaluations for yearling weight gain was positively correlated with genetic evaluations for milk, fat, protein production, and cow survival but negatively correlated with female fertility. Yearling weight gain was also correlated with the direct effect on dystocia, and increased yearling weight gain resulted in greater frequency of dystocia. Of the 1749 Israeli Holstein bulls genotyped with reliabilities >50%, 1445 had genetic evaluations. As genotyping of these bulls was performed using several single nucleotide polymorhphism (SNP) chip platforms, we included only those markers that were genotyped in >90% of the tested cohort. A total of 40,498 SNPs were retained. More than 400 markers had significant effects after permutation and correction for multiple testing (*p*_nominal_ < 1 × 10^−8^). Considering all SNPs simultaneously, 0.69 of variance among the sires’ transmitting ability was explained. There were 24 markers with coefficients of determination for yearling weight gain >0.04. One marker, BTA-75458-no-rs on chromosome 5, explained ≈6% of the variance among the estimated breeding values for yearling weight gain. ARS-BFGL-NGS-39379 had the fifth largest coefficient of determination in the current study and was also found to have a significant effect on weight at an age of 13–14 months in a previous study on Holsteins. Significant genomic effects on yearling weight gain were mainly associated with milk production quantitative trait loci, specifically with kappa casein metabolism.

## 1. Introduction

Numerous studies have considered the economic consequences of animal size for dairy cows (reviewed by [1]). Most studies have concluded that increased cow size has a negative effect on profitability, and several countries have included negative weights for various measures of cow size in selection indices [2]. However, the economic value for growth rate may also be positive for countries in which meat production of surplus calves from the dairy herd is economically important. Genetic and environmental correlations between growth rate and other economic traits have also been computed in various studies, and these generally tend to be economically negative or negligible [1,3].

Brothersone et al. [1] wrote in relationship to the UK Holstein population: “It is unlikely that routine weighing (or type classification) of young stock would be implemented in the national population due to both the cost and the practical problems associated with such a process.” Since the beginning of the 1990s, a large number of Israeli Holstein herds have routinely weighed both male and female calves several times prior to slaughter or calving. Weller and Ezra [3] used 285,800 records on 105,935 animals from 458 herds recorded between 1994 and 2007 to estimate variance components of growth rate for male and female calves, and both sexes jointly. They also estimated genetic and phenotypic trends for growth rate and the genetic correlations between growth rate and other economic traits.

Several studies have recently performed genome-wide association studies on calf weights for both dairy and beef cattle, but most of these analyzes were based on several thousand genotyped cows, e.g., [4]. Considering the large number of markers analyzed, nominal significance levels of 5 or 1% per individual marker are meaningless. After correction for multiple comparisons, only marginally significant effects were found. An exception was the study of Mao et al. [5], but they only considered slaughter records of males with genetic evaluations.

The objectives of this study were to estimate genetic and environmental parameters of yearling growth rate for both male and female Israeli Holstein calves, to compute genetic evaluations for this trait based on the individual animal model, to estimate phenotypic and genetic trends, and to perform a genome-wide association study (GWAS) based on the genetic evaluations of bulls for yearling growth rate with genotypes. Finally, genomic locations with significant effects were compared to the effects found for these locations in previous studies and the effects associated with these markers on other economic traits in cattle.

## 2. Materials and Methods

### 2.1. Data and the Traits Analyzed

The data were 898,014 weight records of Israeli Holsteins collected at commercial farms from January 1994 through February 2021. Records of calves resulting from multiple births were deleted. In addition, records of calves with unknown sire or dam, and weights prior to age 150 days or after 500 days were deleted because an adjustment of weight to the age of one year would not be reliable at these ages. Weller and Ezra [3] estimated genetic parameters for two traits, age corrected weight, and weight gain to the age of one-year yearling weight gain (YG), but the genetic correlation between these traits was nearly complete. Therefore, in this analysis, we only analyzed YG computed as follows:YG = 365 × ((*w* − 35)/*a*) + 35(1)
where *w* = weight in kg at age *a* in days. This formula assumes a birth weight of 35 kg and includes birth weight in the calculation of YG. This value was found be the mean birth weight for Israeli Holsteins [3]. Records with YG values <150 and >650 were deleted because these values probably are the result of recording mistakes. For animals with more than five valid weight records, the first four and the last record up to 500 days were retained. After these edits, the data set consisted of 814,729 records on 368,255 animals recorded in 740 herds. This is denoted “data set 1”, and details are given in Table 1. Of these calves, 162,081 were males and 206,174 were females. Parents and grandparents of the animals with records are also included in the genetic analyses of this data set, and the number of ancestors is also given in Table 1. The number of animals by number of weight records per animal are given in Table 2. Over 60% of the calves had more than one record.

### 2.2. Statistical Analysis

A subset of this data, consisting of records recorded from 1 January 2008 through 31 December 2017, was used to estimate REML variance components for YG, corrected for age at weight. This was denoted “data set 2”, and details are also given in Table 1. Variance and covariance components were computed by the AIREMLf90 program [6] using a single-trait individual animal model. The analysis model based on [3] was as follows:*T_ijkl_* = *G_i_* + *A_j_* + *P_j_* + *H_k_* + *a*^0.5^ + *a* + *a^2^* + *e_ijkl_*(2)
where *T_ijkl_* = the *l*th YG record of calf j from herd k; *G_i_* = the *i*th genetic group effect for animals with unknown parents; *A_j_* = the additive genetic effect of calf j; *P_j_* = the permanent environmental effect of calf j; *H_k_* = the effect of herd-year-season (HYS) k; *a^0^*^.5^, *a*, and *a*^2^ are the square root, linear, and quadratic effects of age; and *e_ijkl_* = the random residual effect. The square root, linear, and quadratic effects of age were all significant (*p* < 0.001) in [3] and were therefore included in the current analysis. Age, G, and HYS effects were fixed, and the other effects were random. Two genetic groups were defined depending on which parents were unknown: group 1 for animals with only the dam unknown and group 2 for animals with sire or both parents unknown. Two seasons were defined for each herd-year relative to date of birth: from April through September and from October through March. Separate HYSs were defined for male and female calves. Therefore, a sex-of-calf effect was not included in the analysis model. Heritability was defined as the *A* variance component divided by the sum of the *A*, *P*, and *e* variance components. Repeatability was defined as the sum of the *A* and *P* variance components divided by the sum of the *A*, *P*, and *e* variance components. Data set 2 was analyzed including calves of both sexes and of males and females separately. Prior to analysis of both sexes, the records of females were multiplied by the square root of the ratio of the male and female additive genetic variance components from the individual sex analyses to bring the records of both sexes to equal genetic variances.

Genetic evaluations for YG were computed for all animals included in data set 1 by the same model as data set 2, except that genetic groups for animals with missing parents were defined by sex of animal, birth year, and which parents were unknown. Although the Israeli dairy cattle population is 99% Holstein, a small fraction of cows was also mated to other bulls, and additional groups were determined by breed of sire for breeds other than Holstein. A total of 64 genetic groups were defined. As in the analysis of data set 2; G, HYS, and age effects were fixed; the other effects were random; and separate HYSs were defined for male and female calves. Although this precluded the need to include a sex-of-calf effect, it does not correct for the fact that variance components were also different by sex. To correct for this, we applied the procedure of Weller and Ezra [3].

For male calves, *A* and *P* variance components as derived from the REML analysis were calculated relative to the residual variance component.Records of female calves were multiplied by the square root of the ratio of the genetic variances between male and female calves. Thus, the additive genetic variance component is now equal for both sexes, and the *P* and residual variances for females are changed by this ratio.The mixed model equations are then constructed with different *P* variances for each sex. For males, the diagonal elements are augmented by the ratio of the *P* and residual effects. For females, the diagonal elements are augmented by (*P_f_* × *A_m_*)/(*A_f_* × *R_m_*), where *P_f_* = the *P* variance for females, *A_m_* = *A* variance for males, *A_f_* = *A* variance for females, and *R_m_* = residual variance for males.Although the residual variance for males is absorbed from the mixed model equations, the residual variance for females is not. The corrected residual variance for records of females is then computed as: (*R_f_* × *A_m_*)/(*A_f_* × *R_m_*), where *R_f_* = the residual variance for females and the other terms are as defined previously. All records of females are then multiplied by the inverse of this ratio in the mixed model equations.

The genetic base for all evaluations was the mean of calves born in 2015. Genetic trends were computed from data set 1 as the regression of estimated breeding values (EBV) of calves born since 1 January 1992, on their birth dates. Phenotypic trends for YG were computed based on the weight records closest to 365 days as the regression of YG on the birth dates of animals with records born since 1992. The reliabilities of the EBV were estimated using the algorithm of Misztal and Wiggans [7], as corrected by Misztal et al. [8].

The correlations were computed among the single sex evaluations of male ancestors derived from the analysis of data set 2 and the combined sex evaluations from data set 1, for bulls with reliabilities >0.9 in the analysis of data set 1. The correlations were also computed between sire EBV with reliabilities >0.5 for YG; EBV for all traits included in the Israeli breeding index, PD16; maternal and direct effects on calving traits; and 17 conformation traits. Genetic evaluations for milk, fat, protein, somatic cell score (SCS), fertility, and persistency were derived by the multitrait animal model as described [9,10]. Genetic evaluations for fat and protein percentage were derived from the genetic evaluations of milk, fat, and protein as described by [9]. Genetic evaluations for 17 conformation traits and herd life were computed by single-trait animal models [11]. Genetic evaluations for first parity dystocia and calf mortality were compute by single-trait sire and maternal grandsire models, as described by [12]. The component traits in PD16 and their index coefficients are given in [13].

### 2.3. Genomic Analysis

The genomic analysis included all Israeli Holstein bulls with genotypes born from 1991 with reliabilities >0.5 from the animal model analysis of data set 1. Of the 1749 Israeli Holstein bulls genotyped, 1445 had genetic evaluations with reliabilities >0.5. As genotyping of these bulls was performed using several SNP chip platforms, we included only those markers that were genotyped in >90% of the tested cohort. A total of 40,498 SNPs were retained. GWAS were computed as described in [13,14]. The response variable was the sires’ transmitting abilities for YG. The additive substitution effects, the coefficients of determination, and the nominal probabilities for the hypothesis of no effect were computed using PLINK software using the –assoc flag for the association test of quantitative traits [15]. This function uses a standard linear regression of phenotype on allele dosage. To control for the nonindependence of individuals within the same family, we generated one million permutations of genotype data against the sires’ transmitting abilties (the phenotype). Finally, a multiple-test correction based on Bonferroni correction was made as detailed in [15]. Thus, the minimal genome-wide probability was <10^−6^ if the substitution effect obtained from the actual data was greater than the permutation effects. To assess the variance explained by all SNPs, we used GCTA-GREML software [16]. The genetic relatedness matrix was calculated using the –make-grm flag, and the variance component calculated using the –grm and the –reml flags.

We obtained the annotation file containing the quantitative trait locus (QTL) database for cattle from the Animal QTLdb (https://www.animalgenome.org/cgi-bin/QTLdb/BT/download?file=gffUMD3.1, accessed on 21 March 2021), and the gtf file with the annotated bovine genome from Ensembl (ftp://ftp.ensembl.org/pub/release-94/gtf/bos_taurus/, accessed on 21 March 2021). Both files are based on the bovine reference genome assembly UMD 3.1, corresponding to this study’s markers coordinates. The GALLO package [17] was applied to annotate the QTLs identified in the GWAS, to find enrichment to previous identified QTLs, and to obtain the genes spanning the QTLs. Gene enrichment analysis was performed using the GeneAnalytics server, which can identify gene enrichment for several terms and data sources, including diseases, pathways, GO terms, and tissue expression [18].

## 3. Results

Means and standard deviations for YG are given in Table 3 by sex. As expected, both means and standard deviations are greater for males. Square root, linear, and quadratic effects of age on YG computed separately for each sex in data sets 1 and 2 are given in Table 4. Although the absolute values of the square root effects were largest and the quadratic effects were smallest, the values were not very similar for the two sexes and data sets. Neither analysis computed the standard errors of these effects.

The REML estimates of variance components, heritability, and repeatability for age-corrected calf weight and YG computed for each sex separately and for both sexes jointly are also given in Table 3. Although variance components and repeatability were higher for males, heritability was higher for females. The variance components for the combined sex analysis were very similar to the values for females. This reflects the fact that there were more female than male records and that the genetic correlation between the sexes is high. The heritability and reliability of the combined evaluation were both between the values for the single sex evaluations.

Mean annual YG derived from the record closest to age 365 days and mean annual EBV for the YG of each animal in data set 1 are plotted in Figure 1 by birth year and sex. With respect to the phenotypic records, a positive trend is evident for males and a negative trend is evident for females. Male weights are 100 to 130 kg greater. Including both sexes, the regression of YG on birth date was −0.96 kg/year, in correspondence with the fact that there were more females than males. A nearly equal positive genetic trend is evident for both sexes; thus, genetically corrected YG increased since 1993. The overall genetic trend was 1.02 kg/year. This genetic trend is considerably higher than the genetic trend of 0.16 kg/year found previously by Weller and Ezra [3].

Correlations among genetic evaluations of 487 bulls for yearling weight gain with reliabilities >0.9 in the analysis of data set 1 are given in Table 5. The correlation between the male and female EBVs based on data set 2 was 0.606, but these evaluations were based on two completely different sets of weight records. Therefore, the correlation between EBV underestimates the genetic correlation. The correlations between the separate sex evaluations of data set 2 and the combined sex evaluations of data set 1 were both higher, but data set 2 was a subset of data set 1. Thus, as assumed previously [3], computing genetic evaluations including both sexes is justified.

Relative contributions of the economic traits to the Israeli breeding index, PD16, and correlations between EBV for YG, PD16, and the major economic traits for Israeli Holstein bulls with reliabilities >0.5 for all traits are given in Table 6. Relative contributions of the calving traits to PD16 and correlations between EBV for YG, and calving traits for bulls with reliabilities >0.5 for YG and the calving traits are given in Table 7. All of the correlations in Table 6 were significant at *p* < 0.0001 except for SCS and milk persistency. All of the significant correlations were in the economically favorable direction with respect to the index traits, except for the correlation with female fertility. Among the calving traits, only the correlation for the direct effect of dystocia was significant and in the economically unfavorable direction. Conformation traits with correlations >0.25 between the EBV for the conformation traits and YG for 1414 bulls with reliabilities >0.5 for both traits are listed in Table 8 by descending order of the magnitude of the correlations. All correlations listed are significant at *p* < 0.0001, and all were in the positive direction. Of the four highest correlations, three were related to mature animal size: body size, stature, and body depth. Conformation traits are not included in the Israeli breeding index, but there is some selection of bull dams based on conformation.

In the previous analysis of this population, genetic correlations with the milk production traits were all positive but below 0.35. The genetic correlations with female fertility and herd-life were both negative, and the correlation with the direct effect of dystocia was economically negative [3]. Thus, only the correlation with herd life is in opposite directions in the two studies. All of the conformation traits with correlations >0.25 were significant in [3], except for udder score and rump width.

The Manhattan plot for the GWAS results for YG is given in Figure 2. There were more than 400 significant markers after permutation and correction for multiple testing (*p*_nominal_ < 1 × 10^−8^). By application of the GCTA-GREML software [16], considering all SNPs simultaneously, 0.69 of variance among the sires’ transmitting ability was explained.

All markers with coefficients of determination >0.04 for YG are presented in Table 9. BTA-75458-no-rs on chromosome 5 explained ≈6% of the variance for the bulls’ transmitting ability for YG. This marker is located ≈10 kb upstream of the gene *SCO2* (Synthesis of Cytochrome C Oxidase 2). This gene is highly expressed in human and cattle adipose tissues, and downregulation of this gene is associated with fat gain and increased insulin resistance [19].

We investigated the association between the YG significant markers and previously reported cattle QTL. The results are summarized in Figure 3. The significant YG SNPs are mainly associated with milk production QTLs, specifically with kappa casein metabolism. To better interpret the association between the identified genomic markers and other QTLs, we performed an enrichment analysis. The number of QTLs annotated within the candidate loci for each trait was compared to the observed number of QTLs in the reference database. The significant enrichments are given in Figure 4. Milk-associated traits are significantly enriched with YG putative QTL, but enrichment was also observed for the traits “average daily gain” and “body weight” at birth and adulthood. This result suggests a possible overlap between milk and protein yield to growth rate pathways. These findings correspond to the relatively high correlation of 0.48 between the EBV of YG and protein yield. Typically, the confidence interval for each significant marker harbors multiple genes, most of which are unrelated to the trait tested. However, assuming that some of the genes that affect the tested trait are part of the same biological pathways, we can expect them to be more frequent among the QTL genes than expected by chance (i.e., their fraction among the QTL genes will be higher than the fraction of the pathway’s genes among all genes). Thus, by identifying these pathways, we can point to specific genes involved in the examined trait. Therefore, we performed an enrichment analysis of the genes spanning the significant markers (+/− 100 kb) with multiple biological terms (i.e., pathways, go terms, and diseases). This analysis revealed significant enrichment in the “Development FGFR Signaling” pathway and the “Calcium Ion Transmembrane Transport” GO term (Appendix A).

## 4. Discussion

As most previous studies show, heritability for YG is within the range of 0.25 to 0.4 [3,20]. Standard errors for both heritabilities and reliabilities were in the range of 0.01 to 0.02. Standard errors for growth rate from previous studies were in the range of 0.03 to 0.05 for smaller samples [1,21].

In 2008, Maher [22] summarized the results of three groups of researchers that performed some of the first GWAS studies for human height. Although huge populations were analyzed and human height has a heritability of 0.9, all the effects found were very small. Altogether, the 40 largest effects accounted for a little more than 5% of height’s heritability. Various explanations were presented for this anomaly. The currently most widely accepted explanation is that the infinitesimal model of Fisher is basically correct [23]. That is, genetic variation in quantitative traits is due to a very large number of factors, each with very small effects [16]. This does not seem to be the case for domestic animals, which have been under intense selection for several generations and have very small effective population sizes. Currently, 160,659 QTL associations from 1030 publications with nominally significant effects for quantitative traits in cattle are recorded in the Animal QTLdb (https://www.animalgenome.org/cgi-bin/QTLdb/BT/index, accessed on 21 March 2021), although the vast majority have not been confirmed on independent studies. Among those associations that have been confirmed by several independent studies, the causative polymorphism has been determined in only a few cases [24].

Compared to other economic traits that have been analyzed by GWAS in commercial animal populations, YG in dairy cattle is somewhat unique in that this trait has relatively high heritability but has not been under intensive selection in dairy cattle.

Although 24 markers with coefficients of determination >0.04 were found, some of these are closely linked and most likely have detected the same causative polymorphism. With respect to independent confirmation of our results, we consider only studies that analyzed growth traits on the major dairy breeds, including Holsteins. None of the 24 markers listed in Table 8 were also listed in the 10 markers flagged for Holsteins in the study by Mao et al. [5], although the traits analyzed in the previous study were based on carcass weight at the slaughter of male calves. Yin and König [4] found a significant effect for ARS-BFGL-NGS-39379, and two additional closely linked markers on chromosome 5 for weight at 13–14 months of age. This marker had the fifth-largest coefficient of determination in the current study. The two markers associated with large effects near the beginning of chromosome 14, between 7 and 8 Mbp, correspond closely to the effect found by [25] for stature and body depth of mature US Holstein cows between 8 and 9 Mbp. Thus, there is a degree of correspondence between the results found in the current and previous studies. The marker BTA-75458-no-rs that had the highest coefficient of determination, ≈6%, is located only 10 kb upstream of the gene *SCO2*. *SCO2* is one of the mitochondrial COX assembly factors. A previous study showed that reduction in *SCO2* activity leads to increased fat mass, adipogenesis, and insulin resistance [19]. It is, therefore, possible that variation in or near *SCO2* regulates its activity and might contribute to the phenotypic variation in the YG.

GWAS often provides multiple QTLs that harbor numerous genes. It is therefore challenging to infer the actual genes and polymorphism that contribute to the phenotypic variation. To address this issue, we performed an enrichment analysis for the genes spanning the marker positions. If the actual genes involved in the tested traits are part of the same pathways, we expect them to be represented among the QTLs genes more than by chance. This analysis revealed significant enrichment of the FGFR pathway (Appendix A). The FGFR genes were shown to participate in energy metabolism regulation, in the embryo and postnatal development and growth, and in fat biogenesis [26,27]. Thus, we propose that polymorphism in or near FGFR genes in the QTLs discovered herein possibly affect YG phenotypic variation.

Although YG is positively correlated with all three milk production traits and longevity, YG is also genetically correlated with mature cow size and smaller cows require less feed for maintenance. Therefore, a case can be made for application of some selection pressure to reduce YG considering that YG has a positive genetic trend and has economically unfavorable genetic correlations with female fertility and the direct effect of dystocia.

## 5. Conclusions

YG of male and female calves is highly correlated genetically; thus, records from both sexes can be combined into a joint genetic analysis. The genetic trend for YG in the Israeli Holstein population was 1 kg/year. YG is positively correlated with milk production traits but economically negatively correlated with fertility and the direct effect of dystocia. In the genome wide association study, >400 markers were significant (*p*_nominal_ < 1 × 10^−8^) after correction for multiple testing and 24 markers had coefficients of determination >0.04. Considering all SNPs simultaneously, 0.69 of variance among the sires’ transmitting ability was explained. The growth rate QTLs are mainly co-associated with milk production QTLs, specifically with kappa casein metabolism. ARS-BFGL-NGS-39379 had the fifth-largest coefficient of determination in the current study and was also found to have a significant effect on weight at age 13–14 months in a previous study on Holsteins. Negative selection on YG as part of a properly weighted selection index may be justified.

## Figures and Tables

**Figure 1 genes-12-00708-f001:**
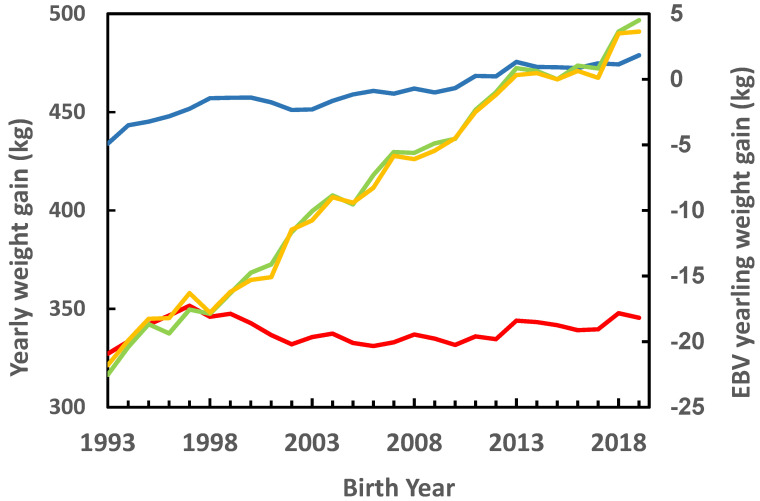
Mean annual yearling weight gains (YG) derived from the record closest to age 365 days of each animal, and mean annual EBV for YG by birth year and sex from data set 1. 
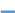
, YG males; 
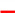
, YG females; 
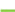
, EBV males; 
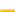
, EBV females.

**Figure 2 genes-12-00708-f002:**
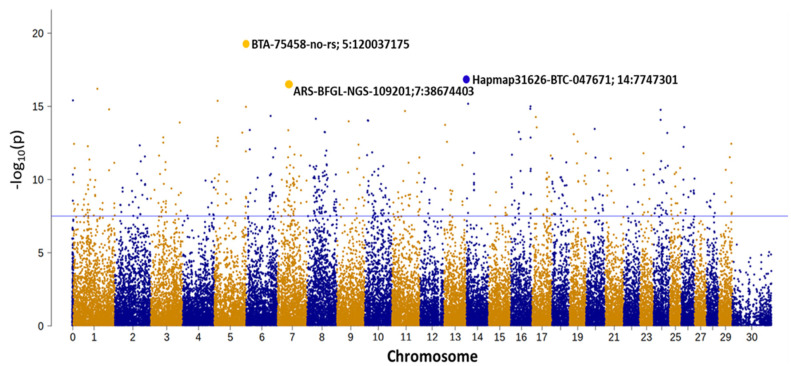
Genome-wide association study Manhattan plot for yearling weight gain. Chromosomal positions are on the *x*-axis, and nominal −log10 *p*-values are on the *y*-axis. Chromosome 0 denotes markers with unknown map positions, and chromosome 30 is the sex chromosome. The horizontal line denotes the genome-wide significance threshold of 0.05, as derived from one million data permutations and correction for multiple testing.

**Figure 3 genes-12-00708-f003:**
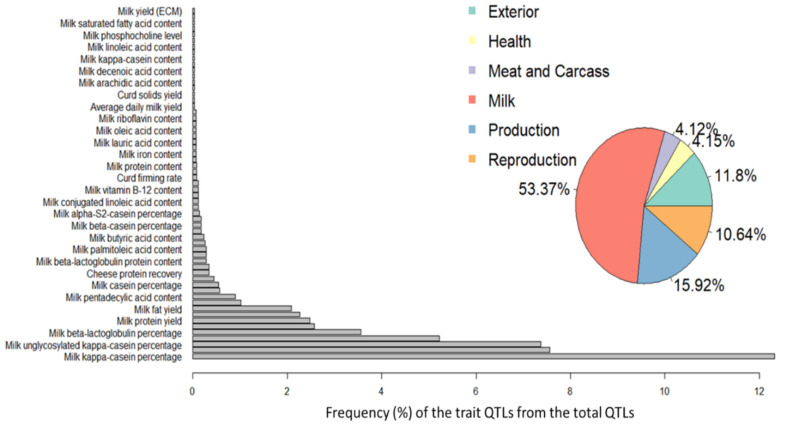
Association between the yearling weight gain (YG) QTLs and previously reported QTLs. Percentage of major class QTLs associated with YG are presented in the pie chart. Distribution of milk production traits QTLs associated with the YG QTLs are presented in the bar plots.

**Figure 4 genes-12-00708-f004:**
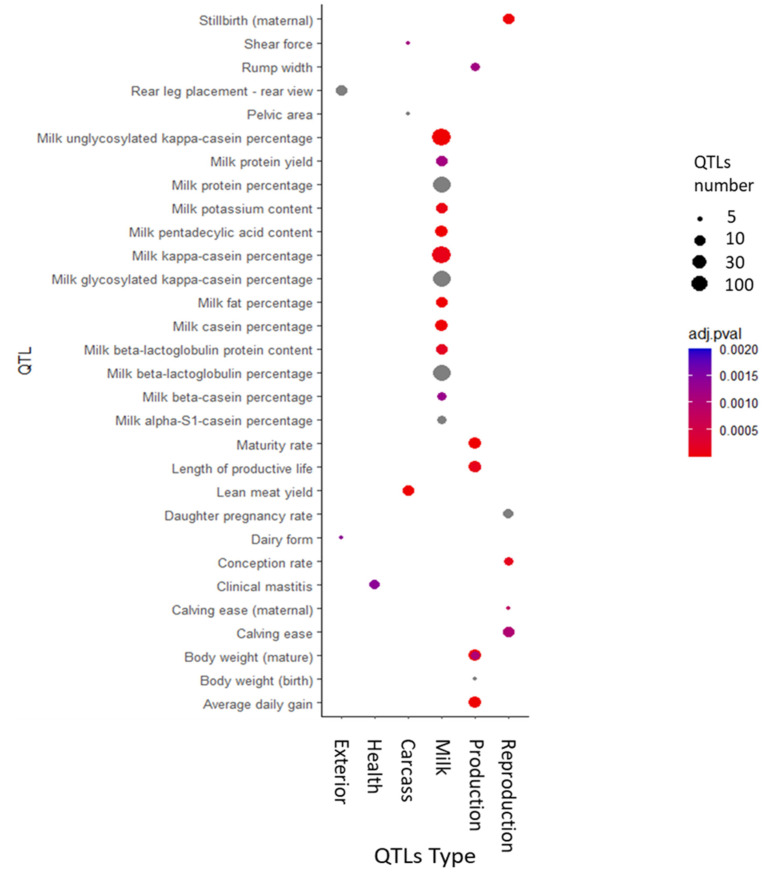
Bubble plot displaying the enrichment QTLs previously reported with the identified yearling weight gain QTLs. The darker the red shade in the circles, the more significant the enrichment. The area of the circles is proportional to the number of QTLs. The *Y*-axis represents the QTL name, and the *X*-axis denotes the affiliation of the QTL with the major trait class.

**Table 1 genes-12-00708-t001:** Number of records and levels of effects in data sets 1 and 2. Data set 1 was the complete data set used to compute genetic evaluations and genetic trends. Data set 2 was used to estimate variance components.

Data Set	Number of:	Males	Females	Both
1	Records	434,639	380,090	814,729
	Animals with records	162,081	206,174	368,255
	Ancestors without records	3019	222,260	225,279
	Herd-year-seasons	−	−	14,523
	Genetic groups	−	−	64
2	Records	152,392	166,361	318,753
	Animals with records	53,013	98,976	151,989
	Ancestors without records	1641	113,377	115,018
	Herd-year-seasons	1083	4512	5595
	Genetic groups	2	2	2

**Table 2 genes-12-00708-t002:** Number of animals by number of weight records per animal in data set 1.

Number of Records per Animal	Number of Animals
1	368,255
2	230,725
3	104,266
4	66,762
5	44,721
Total records	814,729

**Table 3 genes-12-00708-t003:** Means and standard deviations of yearling weight gain by sex of calf and REML estimates of variance components, heritability, and repeatability (± standard errors) computed from data set 2.

	Sex of Calves Analyzed		
	Males	Females	Both
Means	461	337	
Standard deviations	54.4	45.9	
Variance components			
Permanent environment	1030 ± 27	407 ± 11	622 ± 11
Genetic	574 ± 36	489 ± 16	591 ± 17
Residual	497 ± 2.2	484 ± 2.6	488 ± 1.7
Total	2102	1380	1701
Heritability ^1^	0.272 ± 0.02	0.355 ± 0.01	0.347 ± 0.01
Repeatability ^2^	0.763 ± 0.02	0.649 ± 0.01	0.713 ± 0.01

^1^ Genetic variance component divided by total variance. ^2^ Genetic + permanent environment variance components divided by total variance.

**Table 4 genes-12-00708-t004:** Age effects on yearling weight gain.

Data Set	Sex	Age Effects		
		Square Root	Linear	Quadratic
1	Male	15.87	0.293	−0.00117
	Female	−12.40	0.571	−0.00056
2	Male	32.56	−0.429	−0.00078
	Female	36.80	−1.636	0.00068

**Table 5 genes-12-00708-t005:** Correlations among genetic evaluations for 487 bulls between yearling weight gain and reliabilities >0.9 in the analysis of data set 1.

Analysis	Data Set 2	
	Male Calves	Females Calves
Data set 1	0.740	0.812
Data set 2, males		0.606

**Table 6 genes-12-00708-t006:** Relative contributions of the economic traits to PD16, the Israeli breeding index, and the correlations of the bulls’ EBV for yearling weight gain with PD16 and the main economic traits. Correlations are based on 1510 bulls with reliabilities >0.5 for yearling weight gain.

Trait	Relative Contribution to PD16	Correlation
PD16	1	0.411 **
Milk	0	0.385 **
Fat	0.212	0.417 **
Protein	0.373	0.489 **
SCS ^1^	0.110	−0.063 *
Female fertility	0.145	−0.114 **
Herd life	0.096	0.172 **
Milk lactation persistency	0.042	−0.012

^1^ Somatic cell score, negative values are economically favorable. *, significant *p* < 0.05; **, significant *p* < 0.0001.

**Table 7 genes-12-00708-t007:** Relative contributions of the calving traits to PD16 and correlations between the bulls’ EBV for yearling weight gain and calving traits for bulls with reliabilities >0.5 for both traits (negative calving trait values are economically favorable).

Trait	Number of Bulls	Relative Contribution to PD16	Correlation
Dystocia, maternal	1226	0.013	0.024
Stillbirth, maternal	1226	0.010	0.035
Dystocia, direct	556	0	0.198 *
Stillbirth, direct	556	0	−0.079

*, significant *p* < 0.0001.

**Table 8 genes-12-00708-t008:** Correlations between the bulls’ EBV for the conformation traits and yearling weight gain in descending order for 1414 bulls with reliabilities >0.5 for all traits. Only correlations >0.25 are shown. All correlations listed are significant at *p* < 0.0001.

Trait	Correlation
Body size	0.581
Stature	0.492
Total score	0.473
Body depth	0.441
Dairy character	0.427
Udder score	0.317
Rump width	0.312

**Table 9 genes-12-00708-t009:** Single nucleotide polymorphisms associated with yearling weight gain with coefficients of determination >0.04.

Chromosome	SNP ^1^	BP ^2^	Β ^3^	R^2 4^	*p*-Value ^5^
5	BTA-75458-no-rs	120037175	3.27	0.0574	4.27 × 10^−20^
14	Hapmap31626-BTC-047671	7747301	−4.18	0.0493	1.69 × 10^−17^
7	ARS-BFGL-NGS-109201	38674403	2.93	0.0482	3.9 × 10^−17^
1	Hapmap41804-BTA-24071	91554463	−3.39	0.0476	6.31 × 10^−17^
5	ARS-BFGL-NGS-39379	106269362	−3.29	0.0466	6.26 × 10^−14^
5	ARS-BFGL-NGS-73207	12408591	3.06	0.0454	4.19 × 10^−16^
24	ARS-BFGL-NGS-113760	27506980	−2.84	0.0454	1.73 × 10^−15^
0	BTA-79505-no-rs	2430000	−3.75	0.0453	3.91 × 10^−16^
14	ARS-BFGL-BAC-11513	7428315	−3.52	0.0446	6.65 × 10^−16^
5	ARS-BFGL-NGS-55120	120238450	2.79	0.0442	1.08 × 10^−15^
16	ARS-BFGL-NGS-99802	74999809	2.82	0.0439	1.01 × 10^−15^
16	ARS-BFGL-NGS-15423	74158269	2.92	0.0438	1.47 × 10^−15^
1	BTA-53368-no-rs	136278098	−3.30	0.0434	1.61 × 10^−15^
11	UA-IFASA-8854	49473033	2.42	0.0432	2.09 × 10^−15^
6	ARS-BFGL-NGS-83066	92972074	−3.04	0.0421	4.53 × 10^−15^
10	ARS-BFGL-NGS-117447	13704613	2.76	0.0421	9.5 × 10^−15^
17	ARS-BFGL-NGS-22135	13800376	2.78	0.0418	5.38 × 10^−15^
8	ARS-BFGL-NGS-88701	68010939	2.75	0.0416	5.85 × 10^−14^
9	BTA-10828-no-rs	44951803	−2.81	0.0415	1.05 × 10^−14^
8	ARS-BFGL-NGS-108956	33216307	3.50	0.0414	7.1 × 10^−15^
24	BTA-112410-no-rs	27475390	−2.71	0.0413	8.39 × 10^−15^
10	BTB-00412151	12020216	−2.75	0.0411	9.12 × 10^−15^
3	ARS-BFGL-NGS-105427	110272602	−3.10	0.0407	1.27 × 10^−14^
13	ARS-BFGL-NGS-103379	3764223	2.59	0.0402	1.85 × 10^−14^

^1^ Markers are sorted in descending order of the coefficients of determination. ^2^ Marker coordinate according to the bovine UMD3.1 assembly. ^3^ Substitution effect in units of the sires’ transmitting ability. ^4^ Coefficients of determination (denoting the fraction of the variation in the bulls’ transmitting ability that can be explained by specific QTL). ^5^ Nominal *p*-value from *t*-test.

## Data Availability

Restrictions apply to the availability of these data. The data were obtained from the database of the Israel Cattle Breeders Association (ICBA) and are available from the authors with the permission of ICBA.

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
