# Peer review of "Genetic and Genome-Wide Association Analysis of Yearling Weight Gain in Israel Holstein Dairy Calves"

_genes, 2021, doi:10.3390/genes12050708_

Round 1
Reviewer 1 Report
GENES-1198897
Genetic and Genome-Wide Association Analysis of Growth Rate of Israel Holstein Dairy Calves
Moran Gershoni, et al.
This publication sets out to describe a process for selection of Holsteins for yearling growth by calves. The methodology is fairly straight-forward, using mixed animal model approaches and GWAS applications to seek specific genomic regions of large impact.
General observations.
This is a very well-written manuscript that reports on a well-planned project. However, it seems that this may have been part of a larger study that included other measures of weight. My chief concern with this work is that it shows that there could be considerable change in the YG trait as defined, but to what end? The authors do point out that some selection against the trait might be appropriate. What is completely lacking is any discussion of the other critical pieces of information that are needed to truly make use of YG as defined. The beef industry has long made us of yearling gain as a trait. But most often with also knowing BW at birth. Thus selection for birth weight can be held near constant while selecting for improved YG to reduce the impact on dystocia. Furthermore, yearling gain in the absence of frame size, stature, or body condition score makes recommendations difficult since one can not parse out whether the weight gain is due to lipid deposition or muscle accretion. Particularly at times of parenchymal development, excess fat deposition can impact future milk production, at least phenotypically, if not genetically. This could be related to the negative association with persistency in Table 6.
Specific comments and edits
Page Line(s) Comment
1 2 I would suggest that the title should really be about “Yearling Gain” and not “Growth Rate”. 1) While growth over a full year is technically a rate, I think most would expect growth rate to be in terms of kg/day. As I understand the formula on Page 2, Line 76, YG is really defining YW because the birth weight of 35 kg is added back to the adjusted gain. Granted, it is just a constant, but should really be subtracted out of the YG values reported throughout the manuscript.
2 73 “…and yearling weight gain (YG) defined as weight gain to the age of one year.” Note: incorrectly adding the birth weight would not affect the correlations.
2 76 See above.
2 79 Suggest comma after formula and lowercase w.
3 91 In the table, please use “5” instead of “>4”. You stated that only the first four and the last record were used, which creates a maximum of 5, making “>4” confusing.
3 99 same as above.
4 136 “augmented” instead of “augments”.
4 143 This adjustment approach seem s reasonable.
4 157 “percentage” not “percent”.
5 188 Please indicate in Table 3 which values are weights and which are YG. It seems they are all the same, so probably YG (which are really weights as mentioned above). Was this an artifact of an earlier version perhaps?
5 196 In the table, I would left justify the first column, underline “Variance Components”, and then indent the items from there down. It’s a bit confusing this way.
5 211 Why the first record of each animal and not the more accurate one closest to 365 days?
6 218 This is really confusing. The Y axis label in Figure 1 says “Yearly weight gain”, which would seem to match with the trait described, but the Figure caption says “weights”. Which is it. That might explain why line 215 above says the regression of “weight on birth date was 0.07 kgs/year. That does not seem to match the data in the figure. Taken over 27 years, the total change would be 1.89 kg, and the Figure seems to depict more change over time.
7 222 The total genetic gain over the 27 years does not seem to match the reported genetic trend reported Page 6 Line 222.
7 224-229 Please clarify this argument. It seems the modest correlation of 0.606 between males and females would still argue that this is not genetically the same trait. If memory serves, Allan Robertson indicated a standard of around 0.80 to consider to be the same trait.
8 246 Were EBVs of any measure of size available….BCS, Stature, Birth weight?
10 269 “We investigated…”
12 354 Conclusions are ok. I am still concerned that some could over use the trait. So what you say is correct, but I would also add the caveat that inclusion of the trait should be part of an economic selection index. Maybe “Negative selection on YG as part of a properly weighted selection index may be justified.”
Reviewer 2 Report
The manuscript “Genetic and Genome-Wide Association Analysis of Growth Rate of Israel Holstein Dairy Calves” estimated the genetic parameters for yearling weight gains and identified some candidate SNP/genes related to the trait.
Although, the data used for both genetic parameters and GWAS are sufficient, and the results are sensible. I have some concerns about the selection of the trait and method.
- It is not clear about the selection of only this phenotype for the manuscript, while the authors have many other phenotypes in the resource population. The authors might also explore the importance of the phenotype in the current or future selection index.
- The methods using in GWAS are not clear; the authors cited two papers but did not provide a summary of methods.
- Line 183-186, the authors provided some information regarding human gene expression, which is not clear and needed justification
- Genetic parameters are important sections in methods and results but without a discussion about them.
- The authors should pay more attention to the writing and the spelling.
Line 9: 365 × ((weight – 9 35)/age at weight) + 35
Line 11, 21: Some double space in this line and other lines in the manuscript.
Line 13: Either define REML or remove it
Line 16: it is not really clear what the authors mean by “regression of weight on birth date” and the importance of this result?
Line 18: is a genetic trend for both sexes
Line 21: Why positive should be in “ “?, Are the correlation strong or weak, significant or not?
Line 24: Are all of SNP chip medium? Did the authors perform some imputation? Define SNPs and QTLs before using them as the abbreviation
Line 25: Change SNP to SNPs
Line 27: Why 0.04
Line 28: Did the authors mean phenotypic variation or additive genetic variance for “ variance for the genetic evaluations”?
The authors should be consistent with using Yearling weight gain (line 21) or yearling gain (line 31)
Line 51-53: The authors might summarise what they recorded rather than just presented what they did?
Line 56: Since the authors present several studies, I suggest providing some supporting references.
Line 56-57: it is not clear what the authors mean for multiple comparisons?
The introduction is very short, I suggest the authors explore some about the biology of yearling growth rate and the GWAS, and any genomic prediction has done for these traits or growth traits?
Line 72-74: What did these authors analyze: the genetic parameters or GWAS?
Line 77: Add the multiplier symbol in the formula.
Line 80; Please justify why to remove these records.
Line 103-104: Why did the authors use different forms of ages in the models “the square root, linear, and quadratic effects of age,”
Line 136: is the * mean ×
Line 166: There is no reason for not giving the exact number.
Line 167: Please give a summary of GWAS method0; what did the authors use for the response variable in GWAS?
Line 251: Could the authors provide some information regarding PD16
Line 302-311: I am not sure if the discussion about the missing heritability important here.
Why did the authors not discuss the genetic parameter for the trait?
The names of the genes should be written in Italics
Line 344: FGFR pathway genes, I think the word genes should be removed
Remove dot before reference in line 346.
Besides this pathway, did the authors find other pathways have biological meaning?
Round 2
Reviewer 2 Report
Thank the authors for responding to my comments. Most of my comments have been addressed. I still have some concerns:
1 . As the authors mentioned about the importance of missing heritability, the authors should provide the estimation of the variance of the trait explained by significant SNPs in the current analyses.
- Thank the authors for providing new tables and explaining PD16. Still, the authors should pay more attention to the final proofreading and the version I have containing a piece of messy information in table 7 and 8.
- For the genetic parameter section, the authors should move the paragraph to the discussion part; otherwise, the authors should combine the results and discussion in one section.
- Figure 4. What is chromosome 0, 30 standing for? If chromosome 30 is for the sex chromosome, why did the authors include it?
Minor:
Line 79: It might be helpful if the authors give an explanation why using 35kg in the equation.
Table 6: Is the milk persistency as lactation persistency? Give the footnote define SCS.
Line 356: might change million bp = Mb
Figure 5: The authors might increase the size of the dot so that it will be more visible
Figures 1 and 2 can combine into one figure.
Author Response
Answers to the reviewer’s comments are preceeded by "AU".
1 . As the authors mentioned about the importance of missing heritability, the authors should provide the estimation of the variance of the trait explained by significant SNPs in the current analyses.
AU: This is now given in the abstract, lines27-28; and results, lines 288-289. The methodology is explained in lines 190-193.
Thank the authors for providing new tables and explaining PD16. Still, the authors should pay more attention to the final proofreading and the version I have containing a piece of messy information in table 7 and 8.
AU: Titles to these tables have been corrected.
For the genetic parameter section, the authors should move the paragraph to the discussion part; otherwise, the authors should combine the results and discussion in one section.
AU: This section was moved to the discussion, as requested, lines 334-337.
Figure 4. What is chromosome 0, 30 standing for? If chromosome 30 is for the sex chromosome, why did the authors include it?
AU: These chromosomes are explained in the legend to Figure 2. The sex chromosome was included, because only males were genotyped.
Minor:
Line 79: It might be helpful if the authors give an explanation why using 35kg in the equation.
AU: An explanation is now given in lines 84-85.
Table 6: Is the milk persistency as lactation persistency? Give the footnote define SCS.
AU: Table 6 was corrected as suggested.
Line 356: might change million bp = Mb
AU: Corrected as suggested, lines 371-372.
Figure 5: The authors might increase the size of the dot so that it will be more visible.
AU: This figure, now figure 4 was corrected as suggested.
Figures 1 and 2 can combine into one figure.
AU: Corrected as suggested.
